# Urine-Derived Stem Cell-Secreted Klotho Plays a Crucial Role in the HK-2 Fibrosis Model by Inhibiting the TGF-β Signaling Pathway

**DOI:** 10.3390/ijms23095012

**Published:** 2022-04-30

**Authors:** Sang-Heon Kim, Jeong-Ah Jin, Hyung Joon So, Sung Hoon Lee, Tae-Wook Kang, Jae-Ung Lee, Dae Eun Choi, Jin Young Jeong, Yoon-Kyung Chang, Hyunsu Choi, Youngjun Lee, Young-Kwon Seo, Hong-Ki Lee

**Affiliations:** 1Institute of Cell Biology and Regenerative Medicine, EHLBIO Co., Ltd., Uiwang-si 16006, Korea; ksh_8959@naver.com (S.-H.K.); qlrl937@gmail.com (J.-A.J.); shj048650@gmail.com (H.J.S.); shlee@ehlbio.com (S.H.L.); tppong86@gmail.com (T.-W.K.); lju4815@gmail.com (J.-U.L.); lyj6194@naver.com (Y.L.); 2Department of Medical Biotechnology, Dongguk University, Goyang-si 10326, Korea; 3Department of Nephrology, School of Medicine, Chungnam National University, Daejeon 35015, Korea; daenii123@gmail.com (D.E.C.); spwlsdud@naver.com (J.Y.J.); 4Department of Nephrology, Daejeon St. Mary Hospital, Daejeon 34943, Korea; racer@catholic.ac.kr; 5Clinical Research Institute, Daejeon St. Mary Hospital, Daejeon 34942, Korea; 20110201@cmcnu.or.kr

**Keywords:** chronic kidney disease, urine-derived stem cells, mesenchymal stem cells, renal fibrosis, klotho

## Abstract

Renal fibrosis is an irreversible and progressive process that causes severe dysfunction in chronic kidney disease (CKD). The progression of CKD stages is highly associated with a gradual reduction in serum Klotho levels. We focused on Klotho protein as a key therapeutic factor against CKD. Urine-derived stem cells (UDSCs) have been identified as a novel stem cell source for kidney regeneration and CKD treatment because of their kidney tissue-specific origin. However, the relationship between UDSCs and Klotho in the kidneys is not yet known. In this study, we discovered that UDSCs were stem cells that expressed Klotho protein more strongly than other mesenchymal stem cells (MSCs). UDSCs also suppressed fibrosis by inhibiting transforming growth factor (TGF)-β in HK-2 human renal proximal tubule cells in an in vitro model. Klotho siRNA silencing reduced the TGF-inhibiting ability of UDSCs. Here, we suggest an alternative cell source that can overcome the limitations of MSCs through the synergetic effect of the origin specificity of UDSCs and the anti-fibrotic effect of Klotho.

## 1. Introduction

Chronic kidney disease (CKD) is caused by diabetes, hypertension, and glomerulonephritis, which eventually develops into end-stage renal disease (ESRD), requiring renal replacement therapy. Renal replacement therapy deteriorates the patient’s quality of life and increases the social and economic burden due to increased medical expenses [1,2,3,4,5,6,7]. Over the past decade, the incidence of CKD has been managed through efforts to mitigate risk factors such as hypertension and diabetes [8,9]. However, the prevalence of ESRD in the elderly population is still increasing [10].

Tubulointerstitial fibrosis (TIF) is one of the major pathological symptoms of CKD, and is caused by phenotypic changes in cells. Epithelial–mesenchymal transition (EMT) is known to be involved in renal fibrosis [11], and transforming growth factor-beta (TGF-β) is the most potent fibrogenic factor inducing the EMT process [12]. Epithelial cells lose their cellular polarity and intercellular contact during the EMT process. In addition, epithelial cells undergo the reconstruction of cytoskeletal structures, cell shape changes, and the downregulation of epithelial cell-related phenotypes [13]. EMT contributes to CKD development by differentiating epithelial cells into myofibroblast-like cells and increasing ECM deposition [14,15]. Therefore, the most crucial objective of the CKD treatment strategy is to suppress the progression of EMT and fibrosis.

Recently, studies have been conducted to treat CKD using various types of stem cells [16,17,18,19]. Mesenchymal stem cells (MSCs) have been studied as the most realistic alternative, and have demonstrated some efficacy against kidney injury [20,21,22,23,24]. However, some studies have also reported differences between in vivo and in vitro efficacy [25]. Invasive cell collection and the low intra-renal engraftment rate [26] remain limitations to be overcome in the use of MSCs in the treatment of CKD. 

The kidney is an organ that experiences continuous damage by filtering waste products in the body. In stem cell-based CKD therapy, some studies have noted that cells should not lose their characteristics and maintained efficacy when administered to patients with conditions such as high blood pressure and hyperglycemia [27]. Urine-derived stem cells (UDSCs) can be expected to have a more effective therapeutic potential in CKD environments compared to other stem cells because of their origin specificity. Several previous studies have demonstrated excellent therapeutic efficacy in various kidney disease models of UDSCs [23,24,28,29,30]. However, their exact therapeutic mechanism has not been studied much [18,31].

We investigated the CKD-specific therapeutic factors of UDSCs, which are distinct from other stem cells. Klotho, known as an anti-aging protein, has been reported to be predominantly synthesized in the human brain and kidneys, although it is expressed in multiple organs [32,33,34]. In the kidneys, it is known to be expressed mainly in the proximal and distal tubules. Klotho not only functions as a mediator to maintain the homeostasis of the endocrine system but also acts as a hormone in various organs, and acts as an antioxidant and anti-aging agent [35]. Several previous studies have shown that Klotho could directly inhibit the TGF-β1 signaling pathway in the fibrosis of renal epithelial cells [36]. However, in patients with CKD, the blood concentrations of Klotho are reported to decrease as the disease progresses [37,38,39]. To date, no studies have been conducted on Klotho expression in UDSCs.

## 2. Results

### 2.1. Characterization and Homing Effect of UDSCs

Cell attachment was observed on day 3 after seeding, and colonies formed of rice grain-shaped cells—representing typical UDSC morphology—were generated on day 7 [40,41,42]. The cells proliferated and formed colonies, exhibiting a round shape with a clear border, between days 10 and 15. After the first subculture, the morphology of the UDSCs gradually changed from rice grain shapes to MSC-like spindle-shaped cells (Figure 1A), and maintained their rapid proliferation potential until around 3 weeks (Figure 1B).

The UDSCs were strongly positive for MSC surface markers such as CD73 and CD90. Even though the MSC marker CD105 showed sample-dependent expression, all of the samples showed an expression of over 80%. The renal progenitor cell marker CD133, pericyte marker CD146, and pluripotent stem cell marker SSEA4 were also strongly positive, as expected. The UDSCs were strongly negative for endothelial and hematopoietic markers such as CD31, CD34 and CD45 (Figure 1C).

In order to evaluate the differentiation potential, the UDSCs were differentiated into adipocytes, osteocytes, and chondrocytes using a differentiation medium. The Oil Red O, Alizarin Red S, and Alcian Blue staining results indicated that the UDSCs had a typical stem cell differentiation potential (Figure 1D).

We hypothesized that UDSCs have a great homing effect potential because of their kidney tissue-specific origin. Thus, we performed a homing assay using the ischemia-reperfusion acute kidney injury mouse model. Mice were injected with PKH26-labeled UDSCs, and the localization of PKH26-labeled UDSCs was observed using a microscope. The results showed that the UDSCs were specifically located in the damaged areas of the kidneys (Figure 1E).

### 2.2. Klotho Expression Patterns of UDSCs

We have been conducting ongoing investigations to identify a novel therapeutic factor against kidney disease. We focused on Klotho, an anti-aging protein secreted by the kidneys. Klotho plays a pivotal role in maintaining kidney function and endocrine homeostasis. Several studies clarified that the Klotho gene promoter regions were methylated in Klotho non-expressing cells, while they were not methylated in renal tubular epithelial cells [43]. Therefore, we performed gene and protein analysis to identify the ways in which kidney-originating UDSCs predominated over other tissue-originating stem cells.

We compared the expression level of the Klotho gene in adipose-derived stem cells (ADSCs), bone marrow-derived stem cells (BM-MSCs), and umbilical cord-derived stem cells (UC-MSCs), along with UDSCs. As a result, the klotho gene expression level was about 35 times higher in UDSCs, whereas it was relatively weak in the other stem cells (Figure 2A).

We performed Western blotting to identify the presence and the pattern of Klotho proteins in UDSCs by analyzing cell lysates and supernatants. We discovered that Klotho existed in the full-length Klotho (130 kDa) and soluble Klotho (65 kDa) forms in the UDSCs. The results showed that the pattern of the protein from cell lysates was mainly the expression of 130 kDa Klotho, and the protein from the supernatants mainly showed the expression of 65 kDa Klotho. Both Klotho forms were weakly detected in the other stem cells, as expected (Figure 2B).

Moreover, we quantified the amount of Klotho protein in the cell lysates and supernatants using a Klotho ELISA assay. The Klotho concentration in the cell lysates was 661.42 ± 5.69 pg/mg, and 157.67 ± 2.03 pg/mg in the supernatants. Klotho was not detected in the other stem cells (Figure 2C).

Immunofluorescence staining was performed, and we visualized the Klotho expression in the UDSCs (Figure 2D).

### 2.3. TGF-β1 Induced HK-2 Fibrosis Model

In order to identify the anti-fibrotic effect of UDSCs and their correlation with endogenous Klotho, we established a renal tubular fibrosis in vitro model using HK-2 cells [44,45,46]. HK-2 cells are a renal proximal tubular cell line derived from a normal adult kidneys, and have been commonly used in the in vitro study of renal cellular physiology. We mimicked the fibrosis condition using TGF-β1 treatment (10 ng/mL) for 72 h, and observed morphological changes in the HK-2 cells from cuboidal to spindle shapes (Figure 3A).

At the gene level, TGF-β1 stimulation decreased the gene expression of E-cadherin, the epithelial marker, while it increased the gene expression of N-cadherin, the mesenchymal marker, and Snail and Slug, which are EMT markers. EMT is a reversible process in which epithelial cells undergo defined molecular changes to become mesenchymal cells. The expression of ECM-related gene markers such as matrix metalloproteinase (MMP)-2 and -9, fibronectin, and collagen was increased (Figure 3B).

At the protein level, the expression of TGF-β1, a pro-fibrogenic factor, was increased, whereas the expression of the epithelial marker E-cadherin was significantly decreased. These results imply that HK-2 cells were phenotypically transitioned from epithelial to mesenchymal cells. We could also infer that the epithelial cells differentiated into myofibroblast-like cells by inducing α-SMA protein expression. Fibrosis marker expression was significantly stimulated by TGF-β1 in a dose-dependent manner (Figure 3C).

The basal 130 kDa Klotho expression was relatively higher than the 65 kDa Klotho in HK-2 cells, and the 130 kDa Klotho proteins were significantly decreased after TGF-β1 treatment in a dose-dependent manner (Figure 3D). Therefore, we focused on the relationship between fibrosis and Klotho expression.

### 2.4. Klotho siRNA Transfection Efficiency in UDSCs

We assumed that the anti-fibrotic effect of UDSCs was due to Klotho protein expression. Thus, we performed a Klotho siRNA study to clarify whether its anti-fibrotic function depended upon its presence or absence. We found that the gene and protein expression levels were significantly inhibited in a dose-dependent manner after si-Klotho transfection (Figure 4A,B). We confirmed that the viability of UDSCs was not relatively affected by transfection (Figure 4C), and that si-Klotho transfection strongly suppressed immuno-fluorescence in the UDSCs (Figure 4D). 

### 2.5. UDSCs Suppress Fibrosis in HK-2 Cells via the TGF-β Signaling Pathway

The TGF-β signaling pathway regulates biological features, including cell proliferation, the EMT process, and ECM production through both Smad-dependent (canonical) and independent (non-canonical) ways. TGF-β/Smad pathway, the canonical pathway, regulates various fibrosis-related genes through the translocation of phospho-Smad2/Smad3, which results in ECM production and MMP suppression. The TGF-β/ERK pathway, one of the non-canonical pathways, induces myofibroblast transdifferentiation and proliferation via the phosphorylation of ERK [47,48].

We co-cultured TGF-β1-treated HK-2 cells together with Klotho siRNA-treated or untreated UDSCs (Figure 5A). We compared the anti-fibrotic effect of both UDSC groups through TGF-β/Smad and TGF-β/ERK signaling pathways.

Normal UDSCs effectively downregulated Smad2/3 phosphorylation and the expression of MMP-9 and α-SMA in HK-2 cells as expected, whereas si-Klotho UDSC insignificantly downregulated the TGF-β/Smad pathway (Figure 5B). Additionally, we could infer that normal UDSCs suppressed the EMT process based on the results of E-cadherin upregulation and N-cadherin downregulation. However, Klotho siRNA transfection removed the EMT-suppressive potential from the UDSCs (Figure 5C).

Normal UDSCs effectively downregulated ERK1/2 phosphorylation and the expression of phospho-c-Raf in HK-2 cells, as expected, whereas si-Klotho UDSC insignificantly downregulated the TGF-β/ERK pathway (Figure 6A). Our data suggest that normal UDSCs suppress myofibrogenic differentiation and proliferation.

We also compared the Klotho levels in HK-2 cell lysates after co-culture with Klotho siRNA-treated or untreated UDSCs. Normal UDSCs significantly increased the 130 kDa Klotho expression in the HK-2 cells compared to the si-Klotho UDSCs. The 130 kDa Klotho expression of HK-2 cells was relatively increased after co-culture with normal UDSCs compared to si-Klotho UDSCs. However, since the basal 65 kDa Klotho expression was low, none of the groups—including the positive control—showed significant results. Even though there was no statistical significance, the 65 kDa Klotho expression of HK-2 cells seems to be increased after co-culture with normal UDSCs (Figure 6B).

Our data demonstrate that the presence of Klotho proteins in UDSCs downregulated both the canonical and non-canonical pathways in the HK-2 fibrosis model. Thus, we suggest that UDSC-derived Klotho plays a pivotal role in the anti-fibrotic effect and recovery function (Figure 6C).

## 3. Discussion

CKD is a common chronic disease presenting pathological characteristics such as glomerular dysfunction, the fibrosis of the tubules, and the expansion of the mesangium. Tubular atrophy and interstitial fibrosis are the main factors promoting the progression to ESRD. Because the kidneys have limited regeneration capabilities, kidney fibrosis usually proceeds irreversibly [49,50]. Ultimately, the main goal of CKD treatment is to delay and suppress fibrosis, and to further induce the reconstruction of damaged kidney tissue.

UDSCs exhibit mesenchymal stem cell-like properties after EMT in the kidney, making them different from typical MSCs, contributing to the kidney’s own healing mechanism under normal physiological conditions. UDSCs can solve the limitations of other MSCs because they can be obtained from urine repeatedly without causing any pain or aftereffects in the patients [51]. In addition, from a regenerative point of view, organ-specific stem cells are generally ideal for the treatment of the organs from which they originated. In previous studies, UDSCs could differentiate into the smooth muscle cells, epithelial and endothelial cells, and podocytes that form most kidney tissues, and these cells are known to lead to the development of kidney-like tissue. Therefore, because they originated from the kidneys, it can be expected that UDSCs would be more effective than other stem cells.

Although the therapeutic efficacy of UDSCs for various acute and chronic kidney diseases has been revealed in several studies, the existing studies on the specific mechanisms are insufficient [27]. In this study, we tried to clarify the key factors and mechanism of UDSCs, which are distinct from other stem cells. Klotho has been spotlighted as a potential therapeutic factor for CKD because of its renoprotective potentials, such as fibrosis inhibition and oxidative stress reduction. In the last few years, several strategies using Klotho promoter demethylation, PPAR-γ agonist, vitamin D, and recombinant Klotho protein treatments have been reported to increase endogenous and exogenous Klotho levels [52].

Our results showed that UDSCs had an excellent proliferation rate, and are cells with the characteristics of MSCs (surface antigen expression, differentiation ability). After that, we found that UDSCs were stem cells expressing Klotho, a kidney-specific treatment factor. Under normal kidney conditions, Klotho is a key factor mediating various kidney functions [53]. Recently, studies have reported that the development and progression of CKD were significantly associated with a decrease in Klotho, and that Klotho was not only an early biomarker of CKD but also a potential therapeutic factor for CKD [54,55,56,57]. Our findings suggest that UDSCs can be an excellent alternative to other stem cells for CKD treatment. The confirmation of the presence of Klotho in UDSCs and the culture medium in this experiment supports our suggestion.

In order to mimic the tubular fibrosis process in vitro, HK-2 cells were treated with TGF-β1. TGF-β1 treatment increased the expression of fibrosis-related markers in HK-2 cells and decreased the expression of epithelial cell-related markers. In particular, TGF-β1 significantly reduced the endogenic Klotho expression in HK-2 cells. We tried to determine the way in which Klotho, secreted from UDSCs, affected the fibrosis of HK-2 cells using a co-culture.

The co-culture with Klotho-silenced UDSCs did not effectively suppress the TGF-β/Smad signaling pathway or the EMT process of HK-2 cells compared to normal UDSCs. In addition, the analysis of the TGF/ERK signaling pathways showed that Klotho-silenced UDSCs did not efficiently inhibit the differentiation and proliferation of HK-2 cells into fibroblast-like cells. Conversely, normal UDSCs strongly inhibited the fibrosis of HK-2 cells. Furthermore, we observed a significant increase in the Klotho expression of HK-2 cells after co-culture with normal UDSCs. However, it has not been clearly defined whether the increase of Klotho levels in HK-2 is due to the UDSC-secreted Klotho transport or the endogenous Klotho recovery of HK-2 by itself. Further research is needed in order to address this aspect. 

The above results suggest that Klotho is very pivotal in the anti-fibrotic efficacy of UDSCs against HK-2 cells. A previous study by Doi et al. (2011) [36], which showed that Klotho inhibited TGF-β signaling pathways by directly binding to TGF-β receptors, supports our suggestions.

## 4. Materials and Methods

### 4.1. Isolation and Culture 

The Korean Public Institutional Review Board approved the collection of human urine samples in this study. In total, 200 mL urine was collected individually from healthy donors (*n* = 3) for the isolation of UDSCs. The collected urine was filtered using a 100 μm cell strainer (SPL, Pocheon, Korea) and then centrifuged at 500× *g* for 15 min. After removing the supernatant, the pellet was washed using DPBS (Cytiva, Marlborough, MA, USA) containing 1% antibiotic (Gibco, Waltham, MA, USA) and centrifuged at 500× *g* for 10 min (repeated 3 times). The cell pellets were suspended in growth media composed of KSFM and DMEM/F12 in a 1:1 ratio. The growth medium was supplemented with 5% FBS, 20 ng/mL EGF, 10 ng/mL bFGF, 300 ng/mL Hydrocortisone, 1% ITS-E, and 10 μM Y27632. The cells were seeded on a 24-well plate and cultured for 10–15 days at 37 °C. The growth medium was replaced every 2–3 days, and Y27632 was excluded. Colonies with rice grain-shaped cells were selectively collected and expanded after colony formation. 

### 4.2. Characterization 

UDSCs were stained with an endothelial marker (CD31), hematopoietic markers (CD34, CD45, HLA-DR), mesenchymal stem cell markers (CD44, CD73, CD90, CD105), renal progenitor marker (CD133), embryonic/pluripotent stem cell marker (SSEA4), and pericyte marker (CD146) for the cell surface marker analysis (Appendix A). The cells were analyzed using a flow cytometer (BD, FACS verse), and unstained cells were used as a negative control.

### 4.3. Adipogenic and Osteogenic Differentiation

UDSCs were seeded on a 6-well plate (SPL, Pocheon, Korea) at a density of 10,000 cells/cm^2^. For the adipogenic differentiation, the UDSCs were cultured for 21 days with DMEM supplemented with 10% FBS, 1 µM dexamethasone (Sigma-Aldrich, Burlington, MA, USA), 500 µM 3-isobutyl-1-methylxanthine (Sigma-Aldrich, Burlington, MA, USA), 10 µg/mL insulin (Sigma-Aldrich, Burlington, MA, USA), and 100 µM indomethacin (TCI, Tochigi, Japan). For the osteogenic differentiation, the UDSCs were cultured for 21 days with DMEM supplemented with 10% FBS, 100 nM dexamethasone, 10 mM β-glycerophosphate (Sigma-Aldrich, Burlington, MA, USA) and 50 µM ascorbic acid-2-phosphate (Sigma-Aldrich, Burlington, MA, USA). The adipogenic and osteogenic differentiation medium was changed every 2–3 days. Differentiated cells were fixed at room temperature for 30 min with 4% PFA (Sigma-Aldrich, Burlington, MA, USA) and stained with Oil Red O solution (Sigma-Aldrich, Burlington, MA, USA) and Alizarin Red S solution (Sigma-Aldrich, Burlington, MA, USA).

### 4.4. Chondrogenic Differentiation

UDSCs were seeded in an ultra-low-attachment 96-well plate (Corning, Corning, NY, USA) at a density of 10,000 cells/cm^2^. UDSC spheroids were cultured for 21 days with a StemPro™ chondrogenesis differentiation kit (Gibco, Waltham, MA, USA). The chondrogenic differentiation medium was changed every 2–3 days. Differentiated cell spheroids were embedded into the OCT compound (Sakura Finetek, Torrance, CA, USA) and frozen at −80℃ for cryosection. The cryosections were fixed at room temperature for 30 min with 4% PFA (Sigma-Aldrich, Burlington, MA, USA) and stained with Alcian Blue solution (Sigma-Aldrich, Burlington, MA, USA).

### 4.5. Homing Assay

UDSCs were labeled with PKH26 Red Fluorescent Cell Linker Kit (Sigma-Aldrich, Burlington, MA, USA) according to the manufacturer’s instructions. Bilateral renal clamping was performed for 30 min to induce acute kidney injury in BALB/c *nu/nu* mice (8 weeks). After 24 h of surgery, PKH26-labeled UDSCs were administered intravenously (iv) at 2 × 10^6^ cells/head. The mice were euthanized after 24 h, and the PKH26-labeled UDSCs were observed by microscope.

### 4.6. Cell Culture

HK-2 cells were maintained in DMEM/F12 (Gibco, Waltham, MA, USA) supplemented with 10% FBS. Adipose-derived stem cells, bone marrow-derived stem cells, and umbilical cord-derived stem cells were purchased from Cefobio Inc. (Seoul, Korea). The stem cells were maintained in DMEM supplemented with 10% FBS. The UDSCs’ Klotho expression pattern was compared with multiple samples (*n* = 3) of each stem cell (ADSCs, BM-MSCs, UC-MSCs, and UDSCs).

### 4.7. siRNA Assay

UDSCs were seeded and expanded until they reached 80% confluence. Klotho siRNA was purchased from IDT Inc. (Newark, NJ, USA) and transfected into the UDSCs using TransIT-SiQUEST™ siRNA transfection reagent (Mirus, Marietta, GA, USA). The knockdown efficiency of Klotho was analyzed by real-time PCR and Western blotting after 2 days. Universal negative control siRNA (IDT, Newark, NJ, USA) was used as a non-targeting control.

### 4.8. Co-Culture Assay

The indirect co-culture of UDSCs and HK-2 was performed using a transwell system. Each UDSC and HK-2 was seeded on the upper and lower chamber in a 1:1 ratio (Effector: Target). The medium was replaced with DMEM/F12 containing 0.1%, FBS, 10 ng/mL recombinant human TGF-β1 (Peprotech, Rocky Hill, NJ, USA) to induce fibrosis. The co-culture was maintained for 72 h without a medium change.

### 4.9. Real-Time PCR

The total RNA and cDNA were extracted and synthesized using an RNA extraction kit (Takara, Shiga, Japan) and a cDNA synthesis kit (Takara, Shiga, Japan) according to the manufacturer’s instructions. Real-time PCR was performed with the corresponding primers using a Quant Studio 3 Real-Time PCR Instrument (Applied Biosystem, Waltham, MA, USA) to analyze the expression of mRNA quantitatively (Appendix A). The relative expression of each gene was calculated by the delta–delta Ct method, and is shown in the graph. 

### 4.10. Western Blotting 

Cells were lysed in RIPA buffer (Thermo Fisher Scientific, Waltham, MA, USA) for the extraction of the proteins. The total protein was determined using a the BCA protein assay kit (Thermo Fisher Scientific, Waltham, MA, USA). Equal amounts of protein (20 to 30 μg/lane) were loaded into a polyacrylamide gel (Biorad, Hercules, CA, USA) and transferred to a nitrocellulose membrane (Thermo Fisher Scientific, Waltham, MA, USA). The membranes were blocked with 5% skim milk TBS-T and incubated overnight with the corresponding primary antibodies at 4 °C (Appendix A). Then, the membranes were incubated for 2 h with HRP-conjugated secondary antibodies (1:5000) at room temperature. The target proteins were detected using ECL solution (Thermo Fisher Scientific, Waltham, MA, USA) and analyzed quantitatively with Amersham^TM^ Imager 680 (GE Healthcare, Chicago, IL, USA).

### 4.11. Klotho ELISA Assay

Cells were seeded into a T75 flask at a density of 10,000 cells/cm^2^, and the medium was changed to fresh medium when it reached about 80% confluence. Cell lysates and supernatants were collected after 48 h. A human soluble Klotho ELISA (IBL, Minneapolis, MN, USA) assay was performed in order to quantify the amount of Klotho proteins.

### 4.12. Immunofluorescence Staining

Cells were fixed with 4% PFA (Sigma-Aldrich, Burlington, MA, USA) and permeabilized with 0.25% Triton X-100 (Sigma-Aldrich, Burlington, MA, USA) at room temperature. The fixed cells were blocked with 10% goat serum (Gibco, Waltham, MA, USA) and incubated overnight with the corresponding primary antibodies at 4 °C (Appendix A). Then, the cells were incubated for 2 h with fluorescence-conjugated secondary antibodies (1:5000) at room temperature. The target proteins were detected using a microscope (Nikon, Tokyo, Japan).

### 4.13. Statistical Analysis

The statistical analysis was processed using Graphpad Prism (San Diego, CA, USA). One-way ANOVA was used to perform comparisons of three or more groups, and Student’s *t*-test was used to perform comparisons of two different groups. The data were presented as the mean ± SEM of at least three individual experiments, and values of less than 0.05 were considered statistically significant.

## Figures and Tables

**Figure 1 ijms-23-05012-f001:**
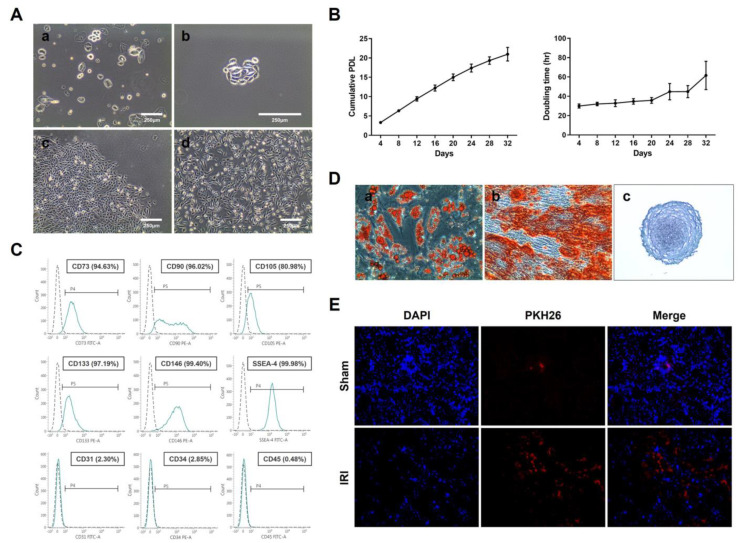
Characterization and homing effect of UDSCs. (**A**) Isolation and culture of UDSC. (**a**) Non-adherent cells in s fresh urine sample (Day 0). (**b**) Cell attachment (Day 3). (**c**) Colony formation of rice grain-shaped cells (Day 10). (**d**) Morphological changes into spindle-shaped cells (Passage 1). (**B**) Representative growth curve of the UDSCs. (**a**) The cumulative population doubling level (CPDL) and (**b**) doubling time (DT) were analyzed at each passage. (**C**) The representative surface marker expression was detected by flow cytometry at passage 5. (**D**) Multilineage differentiation of UDSCs. (**a**) Adipogenic differentiation; Oil Red O staining for lipid droplets (Day 21). (**b**) Osteogenic differentiation; Alizarin Red S staining for calcium deposits (Day 21). (**c**) Chondrogenic differentiation; Alcian Blue staining for glycosaminoglycan accumulation (Day 21). (**E**) PKH26-labeled UDSCs (red) were intravenously administered to wild-type (Sham) and ischemia-reperfusion injury (IRI) mice.

**Figure 2 ijms-23-05012-f002:**
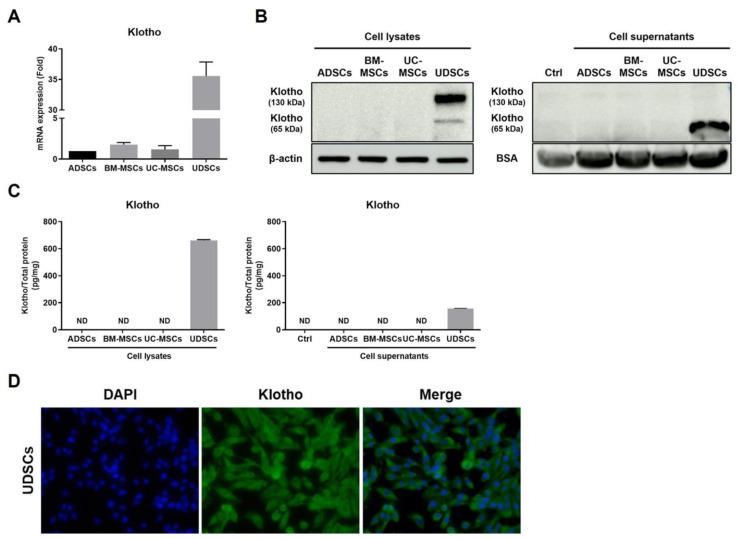
Klotho expression patterns of UDSCs. (**A**) The Klotho gene expression level was analyzed by real-time PCR using cell lysates. (**B**) The Klotho protein expression level was detected by Western blotting using cell lysates and supernatants. (**C**) The Klotho protein concentration was quantified by ELISA using cell lysates and supernatants. The results are shown as the mean ± SEM. (**D**) The Klotho expression was visualized by fluorescence staining (green) in UDSCs.

**Figure 3 ijms-23-05012-f003:**
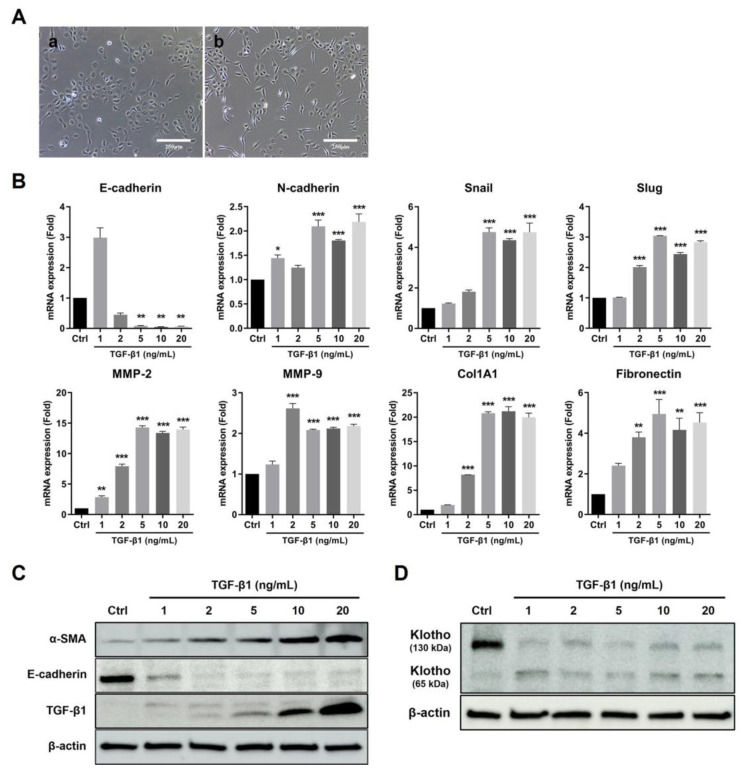
TGF-β1-induced HK-2 fibrosis model. (**A**) HK-2 cells were treated with 10 ng/mL TGF-β1 for 72 h. The morphological changes were observed using a microscope. (**a**) Normal HK-2 cells. (**b**) TGF-β1 treated HK-2 cells. (**B**) Fibrosis-related genes were analyzed by real-time PCR using HK-2 cell lysate. (**C**,**D**) Fibrosis-related proteins and Klotho were detected by Western blotting using HK-2 cell lysate. The results are shown as the mean ± SEM. * *p* < 0.05, ** *p* < 0.01, *** *p* < 0.001 vs. Control.

**Figure 4 ijms-23-05012-f004:**
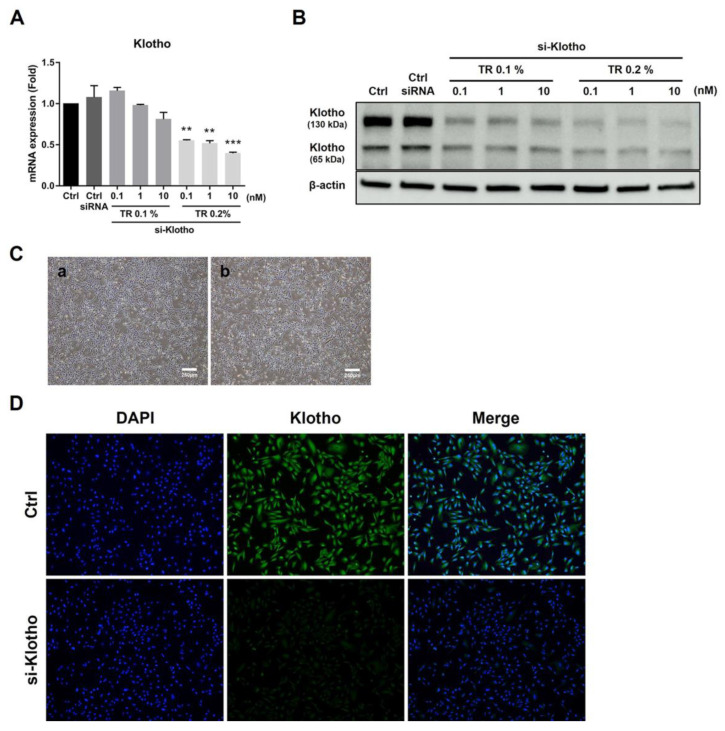
Klotho siRNA transfection efficiency in UDSCs. (**A**,**B**) UDSCs were treated with various concentrations of transfection reagent (TR) and Klotho siRNA. The transfection efficiency was evaluated using real-time PCR and Western blotting. Universal negative siRNA control was used as a transfection control. (**C**) UDSCs were treated with 0.2% transfection reagent and 10 nM Klotho siRNA for 48 h. The morphological changes were observed using a microscope. (**a**) Normal UDSCs, and (**b**) si-Klotho UDSCs. (**D**) Klotho expression was visualized by fluorescence staining (green) in normal and si-Klotho UDSCs. The results are shown as the mean ± SEM. ** *p* < 0.01, *** *p* < 0.001 vs. Control.

**Figure 5 ijms-23-05012-f005:**
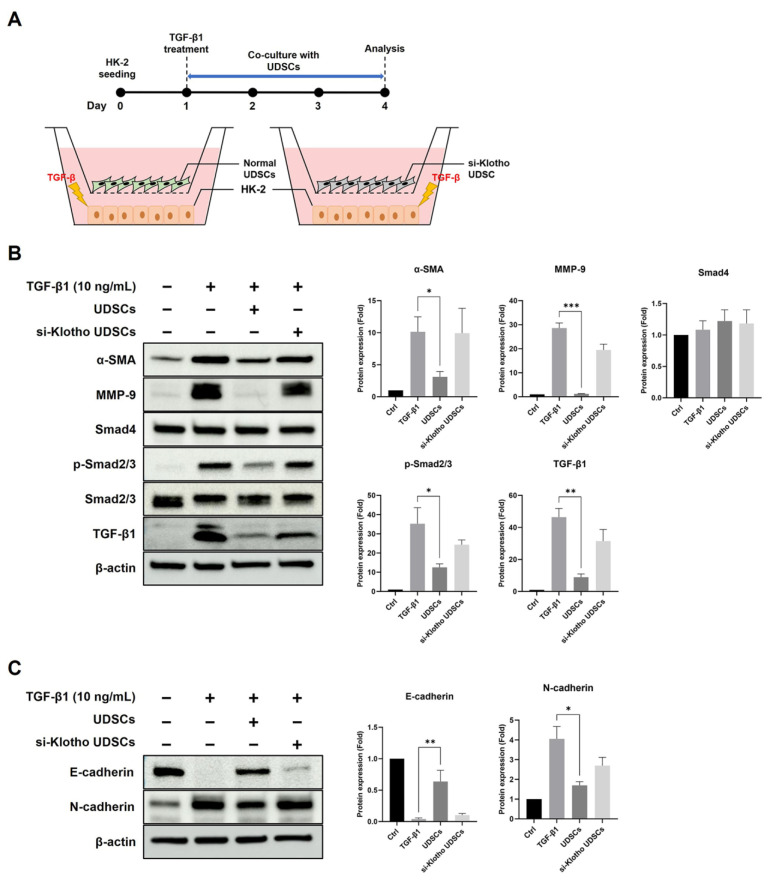
UDSCs suppress fibrosis in HK-2 cells via the TGF-β/Smad signaling pathway. (**A**) The experimental design of an indirect co-culture assay. (**B**) In total, 10 ng/mL TGF- β1-treated HK-2 were co-cultured with normal UDSCs or si-Klotho UDSCs for 72 h. The TGF-β/Smad signaling pathway was analyzed by Western blotting using HK-2 cell lysates. (**C**) EMT-related proteins were analyzed by Western blotting using HK-2 cell lysates. The results are shown as the mean ± SEM. * *p* < 0.05, ** *p* < 0.01, *** *p* < 0.001 vs. Control.

**Figure 6 ijms-23-05012-f006:**
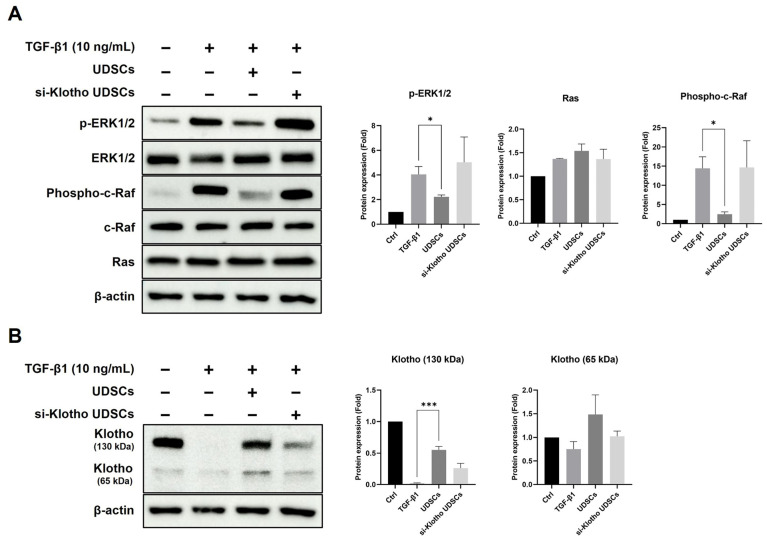
UDSCs suppress fibrosis in HK-2 cells via the TGF-β/ERK signaling pathway. (**A**) In total, 10 ng/mL TGF- β1-treated HK-2 were co-cultured with normal UDSCs or si-Klotho UDSCs for 72 h. The TGF-β/ERK signaling pathway was analyzed by Western blotting using HK-2 cell lysates. (**B**) Klotho expressions were analyzed by Western blotting using HK-2 cell lysates. The results are shown as the mean ± SEM. * *p* < 0.05, *** *p* < 0.001 vs. Control. (**C**) Schematic diagram of the mechanism of UDSC-derived Klotho.

## Data Availability

The data presented in this study are available on request from the corresponding author.

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
