# Peer review of "Urine-Derived Stem Cell-Secreted Klotho Plays a Crucial Role in the HK-2 Fibrosis Model by Inhibiting the TGF-β Signaling Pathway"

_ijms, 2022, doi:10.3390/ijms23095012_

Round 1
Reviewer 1 Report
This manuscript deals with the role of Klotho protein from urine-derived stem cells (UDSCs) in reducing the TGF-beta mediated fibrosis.
The authors have derived and characterised UDSC, and then they have analysed their ability of influencing the TGF-beta induce epithelial mesenchymal transition (EMT). The EMT is determinant in the transformation of epithelial cells into myofibroblast-like cells that are responsible for the extracellular matrix deposition and consequent fibrosis evidenced in chronic kidney disease (CDK).
It appears that Klotho can interact and compete with TGF-beta with TGF-beta receptor in blocking the effect due to TGF-beta as previously reported (doi:10.1074/jbc.M110.174037). The main message of this work is that UDSC can produce Klotho and that, as expected, the silencing of Klotho (and the reduction of production of Klotho) can lead to inhibition of the anti-fibrotic effect of UDSC.
The first point that should be addressed by the author is to state clearly from how many distinct samples (from different donors) UDSC have been derived. The same should be considered for all the experiments performed. Indeed, this is not indicated anywhere in this manuscript. Also, the specific feature of UDSC, that is the expression of high levels of Klotho mRNA and protein, should be compared with multiple samples (at least three different samples) of adipose-derived stem cells (ADSCs), bone marrow-derived stem cells (BM-MSCs), and umbilical cord-derived stem cells (UC-MSCs). This is akey point because the main and novel message of this paper is that UDSC are a source of MSC and of Klotho.
Also, the analysis of Klotho in healthy and CDK patients derived UDSC should be considered. This is a key point to connect the in vitro data with the in vivo fibrosis.
It is relevant to see whether UDSC can produce or not TGF-beta compared with the other MSC used in this work. Indeed, it cannot be excluded that several factors present at the same time can be responsible for the observed effect and differences from the various MSC used.
Another relevant point is to determine the kind of response of UDSC to those conditions that usually lead to CKD, not only the basal/spontaneous production and expression of Klotho that seems to be specific to UDSC.
It is not clear: "Endogenous Klotho expressions were analyzed by Western blotting using HK-2 cell lysates” in the legend to figure 6 panel B.
Indeed, can HK-2 cells express Klotho? It seems so. Thus, how can the authors distinguish from the effect of UDSC-derived Klotho and that from HK-2? Do you mean that UDSC-derived Klotho enter HK-2 cells?
To complete the scenario, does the TGF-beta affect the Klotho production by UDSC (or HK-2)? This point can be of relevance to clarify better the role of these cells in the context of CKD.
Furthermore, the use of soluble Klotho by itself (and the secreted UDSC-derived Klotho) can actually inhibit the HK-2 EMT transformation as expected based on the data shown?
Author Response
Point 1:
The first point that should be addressed by the author is to state clearly from how many distinct samples (from different donors) UDSC have been derived. The same should be considered for all the experiments performed. Indeed, this is not indicated anywhere in this manuscript.
Response 1:
UDSCs were collected from three different donors.
As we mentioned in “4.13. Statistical analysis”, all analyses performed at least three individual experiments.
We revised as below.
- Materials and Methods
(Line 289-290) 200 mL of urine sample was collected individually from healthy donors (n = 3) for UDSCs isolation.
Point 2:
Also, the specific feature of UDSC, that is the expression of high levels of Klotho mRNA and protein, should be compared with multiple samples (at least three different samples) of adipose-derived stem cells (ADSCs), bone marrow-derived stem cells (BM-MSCs), and umbilical cord-derived stem cells (UC-MSCs). This is a key point because the main and novel message of this paper is that UDSC are a source of MSC and of Klotho.
Response 2:
Yes, we performed three individual analyses using multiple samples (n = 3) of each stem cells. We also do have three individual sets of Western blotting images. If we need to attach all of them, please let us know.
We revised as below.
- Materials and Methods
(Line 340-342) UDSCs Klotho expression pattern was compared with multiple samples (n = 3) of each stem cells (ADSCs, BM-MSCs, UC-MSCs, and UDSCs).
Point 3:
Also, the analysis of Klotho in healthy and CKD patients derived UDSC should be considered. This is a key point to connect the in vitro data with the in vivo fibrosis.
Response 3:
We also have considered the differences in Klotho expression between healthy and CKD patients. Thus, we have already analyzed Klotho levels from both groups to support our upcoming clinical trials targeting CKD.
As we expected, UDSCs from CKD patients showed lower Klotho levels than healthy donor (Healthy donor = 740.0 ± 224.7 pg/mg, CKD patient = 376.7 ± 119.5 pg/mg, *p<0.05) .
(But, the data were not shown in the previous manuscript. Because we thought that the theme of this paper specifically focused on the role of Klotho in UDSCs-mediated anti-fibrosis mechanism.)
We attached it as supplementary data.
Supplementary data S1. The Klotho expression levels of UDSCs from healthy donors (n=5) and CKD patients (n=5).
Point 4:
It is relevant to see whether UDSC can produce or not TGF-beta compared with the other MSC used in this work. Indeed, it cannot be excluded that several factors present at the same time can be responsible for the observed effect and differences from the various MSC used.
Response 4:
We analyzed basal TGF-β1 secretion of HK-2, ADSCs, BM-MSCs, UC-MSCs, and UDSCs using TGF-β1 ELISA kit (R&D systems, United States). ADSCs showed predominant TGF-β1 secretion level, but there were no significant differences among the other cells (HK-2 = 383.3 ± 61.7 pg/mL, ADSCs = 1362.7 ± 180.6 pg/mL, BM-MSCs = 442.4 ± 58.8 pg/mL, UC-MSCs = 311.3 ± 17.8 pg/mL, UDSCs = 424.4 ± 61.4 pg/mL).
We attached it as supplementary data.
Supplementary data S2. The basal TGF-β1 secretion levels of various stem cells and HK-2 cells.
(We agree with your point that the cooperative effect of several factors should be considered. But, in this manuscript, we wanted to focus on the different Klotho expression patterns among various stem cells not the differences in the anti-fibrosis effects among various stem cells.)
Point 5:
Another relevant point is to determine the kind of response of UDSC to those conditions that usually lead to CKD, not only the basal/spontaneous production and expression of Klotho that seems to be specific to UDSC.
Response 5:
Fibrosis is one of the major pathological conditions of CKD progression and is triggered by TGF-β activation, the most potent fibrogenic factor. We mimic the fibrosis condition using TGF-β and we analyzed the Klotho expression of UDSCs after treating TGF-β1 for 72 hours using Western blotting. The results showed that the Klotho expressions of UDSCs were decreased in a dose-dependent manner after TGF-β1 treatment.
We attached it as supplementary data.
Supplementary data S3. The Klotho expression of UDSCs after TGF-β1 treatment for 72 hours.
Point 6:
It is not clear: "Endogenous Klotho expressions were analyzed by Western blotting using HK-2 cell lysates” in the legend to figure 6 panel B. Indeed, can HK-2 cells express Klotho? It seems so. Thus, how can the authors distinguish from the effect of UDSC-derived Klotho and that from HK-2? Do you mean that UDSC-derived Klotho enter HK-2 cells?
Response 6:
In Figure 3D, 130 kDa Klotho expression was severely decreased in HK-2 cells after TGF-β1 treatment. And we observed a significant increase in 130 kDa Klotho expression of HK-2 cells after co-culture with normal UDSCs (Figure 6B). Thus, we presumed that 130 kDa full-length Klotho, detected in cell lysates, might be an HK-2-derived membrane Klotho. But we also discussed that the secretion of UDSCs-derived Klotho cannot be excluded as you mentioned.
Therefore, we would like to revise and add the manuscript as below.
- Results
(Line 209-210) Figure 6. Klotho expressions were analyzed by western blotting using HK-2 cell lysates.
- Results
(Line 217-225) We also compared the Klotho levels in HK-2 cell lysates after co-culture with Klotho siRNA-treated or untreated UDSCs. Normal UDSCs significantly increased 130 kDa Klotho expression in the HK-2 cells compared to the si-Klotho UDSCs. 130 kDa Klotho expression of HK-2 cells was relatively increased after co-culture with normal UDSCs compared to si-Klotho UDSCs. However, since the basal 65 kDa Klotho expression was low, all groups including the positive control did not show significant results. Even though there was no statistical significance, 65 kDa Klotho expression of HK-2 cells seems to be increased after co-culture with normal UDSCs (Figure 6B).
- Discussion
(Line 275-280) Conversely, normal UDSCs strongly inhibited the fibrosis of HK-2 cells. Also, we observed a significant increase in Klotho expression of HK-2 cells after co-culture with normal UDSCs. But it has not been clearly defined whether the increase of Klotho levels in HK-2 is due to the UDSCs-secreted Klotho transport or the endogenous Klotho recovery of HK-2 by itself. Further research is needed to address this aspect.
Point 7:
To complete the scenario, does the TGF-beta affect the Klotho production by UDSC (or HK-2)? This point can be of relevance to clarify better the role of these cells in the context of CKD.
Response 7:
As we mentioned above ‘Point 5’, “Supplementary data S3” showed that Klotho expression of UDSCs was expressed in a dose-dependent manner after TGF-β1 treatment.
Point 8:
Furthermore, the use of soluble Klotho by itself (and the secreted UDSC-derived Klotho) can actually inhibit the HK-2 EMT transformation as expected based on the data shown?
Response 8:
There are several references supporting that Klotho protein can inhibit the EMT process in several epithelial cells including HK-2 cells. Especially, Yang L. et al (2021), reported that Klotho prevents EMT process through Egr-1 downregulation in the HK-2 in vitro model and diabetic nephropathy in vivo model. (DOI: 10.1136/bmjdrc-2020-002038)
We also believe that UDSCs-secreted Klotho can inhibit TGF-β1 induced EMT process.
Reviewer 2 Report
This study, written by Dr. Kim SH et al., original research, with the title of “Urine-derived Stem Cell-secreted Klotho Plays a Crucial Role in the HK-2 Fibrosis Model by Inhibiting the TGF-β Signaling Pathwaya” analyzed the Klotho expression in urine-derived stem cells (UDSCs) and the relationship with renal fibrosis in in vitro models. The manuscript is well written, it is clear and easy to read and has enough figures and references. To improve the manuscript, please refer to the following comments:
- A list of abbreviations could be included in the text.
- Lines 54–57. “Urine-derived stem cells (UDSCs), derived from the harsh environment of the kidneys, can be expected to have more effective therapeutic potential in CKD environments compared to other stem cells..” Could you please explain why?
- Lines 61–62. “Klotho, known as an anti-aging protein, has been reported to be predominantly expressed in the human brain and kidneys [32–34].” According to The Human Protein Atlas, Klotho (KL) is also highly expressed in endocrine tissues, urinary bladder, and female tissues. https://www.proteinatlas.org/ENSG00000133116-KL/tissue
- It was stated that the UDSCs had been isolated and cultured from the urine of healthy donors. Could you please explain the procedure for researchers who may not be familiar the specific technique? What is the purpose of using Selective Rho kinase inhibitor? How are the UDSCs separated from the other cells (epithelial, etc)?
- In Figure 1D. The letters “a”, “b”, and “c” are missing.
- Could you please provide more information/explanation for Figure 1E? In the text of Figure 1E, the meaning of IRI could be written (ischemia-reperfusion acute kidney injury). Could you please add the meaning of sham (sham surgery, placebo surgery in the figure’s text?
- Regarding Figure 2B. Could you please explain why B-actin was used in cell lysates and BSA in supernatants? Should the loading control also be present in the western blot image?
- Line 139. It may help readers not familiar with this type of research to explain that HK-2 cells are a proximal tubular cell line derived from a normal human adult male kidney; the cell line has applications in toxicology research.
- Line 142. Instead of Figure 3A, shouldn’t be written 3B?
- Line 150. The authors could add that “EMT is a reversible process where epithelial cells undergo defined molecular changes to become motile mesenchymal cells” to help the readers.
- Lines 182–182. Could you please provide more details about the difference of canonical and non-canonical signaling pathways? Why are they relevant in this situation?
- Line 204. Could you please confirm that instead of “Raf,” “phosphor-c-Raf” should be written?
- Lines 214–217. If I am not wrong, I think that it should be stated that “the presence of Klotho proteins in UDSCs downregulated….”
- Lines 259–260. I think it should be noted the result of experiment 6B in this point of the discussion.
- Could you please discuss the mechanism of Klotho expression by HK2 cells?
- The work of Koh et al. (PMID: 11162628 DOI: 10.1006/bbrc.2000.4226) shows the lack of expression of Klotho in chronic renal failure. If figure 3 could be added to this manuscript, it will help the highlight the importance in human pathological conditions.
- Could you please discuss how to increase Klotho levels and their usefulness? (demethylation of the klotho promoter, diacylation, PPAR-c antagonist, vitamin D, exogenous, etc.). Front. Endocrinol., August 27, 2020 | https://doi.org/10.3389/fendo.2020.00560
Author Response
Point 1:
A list of abbreviations could be included in the text.
Response 1:
We added the list of abbreviations.
Supplementary table S5. List of abbreviations.
Point 2:
Lines 54–57. “Urine-derived stem cells (UDSCs), derived from the harsh environment of the kidneys, can be expected to have more effective therapeutic potential in CKD environments compared to other stem cells.” Could you please explain why?
Response 2:
This sentence could be misunderstood (“harsh environment”) by readers, so we revised it as follows.
- Introduction
(Line 54-57) Urine-derived stem cells (UDSCs) can be expected to have more effective therapeutic potential in CKD environments compared to other stem cells because of their origin specificity.
Point 3:
Lines 61–62. “Klotho, known as an anti-aging protein, has been reported to be predominantly expressed in the human brain and kidneys [32–34].” According to The Human Protein Atlas, Klotho (KL) is also highly expressed in endocrine tissues, urinary bladder, and female tissues. https://www.proteinatlas.org/ENSG00000133116-KL/tissue.
Response 3:
Klotho is predominantly synthesized and secreted by the brain and kidneys in the human body and acts as a hormone that signals with multiple organs. (DOI: 10.1007/978-1-4614-0887-1_9)
Therefore, Klotho expression occurs in various organs.
We revised this sentence as follows to emphasize the synthesis and secretion of Klotho.
- Introduction
(Line 61-63) Klotho, known as an anti-aging protein, has been reported to be predominantly synthesized in the human brain and kidneys although it is expressed in multiple organs[32-34].
Point 4:
It was stated that the UDSCs had been isolated and cultured from the urine of healthy donors. Could you please explain the procedure for researchers who may not be familiar the specific technique? What is the purpose of using Selective Rho kinase inhibitor? How are the UDSCs separated from the other cells (epithelial, etc)?
Response 4:
Y27632 is widely used for stem cell isolation and expansion. In this study, we also added Y27632 to enhance colony-forming potential and viability during UDSC isolation. (DOI: 10.3390/jcm9030827)
As we mentioned in “4.1. Isolation and culture”, first, we seeded urine cells on a 24-well plate at a low density. And then, we selected UDSCs colonies using a microscope. UDSCs have a rice-grain shaped morphology, and UDSCs colonies exhibit a round shape with a clear border (Figure 1A).
The other epithelial cells have significantly lower growth rates than UDSCs. And they generally do not form colonies with a round and clear border like UDSCs colonies. It is a general method for UDSCs isolation.
Point 5:
In Figure 1D. The letters “a”, “b”, and “c” are missing.
Response 5:
We corrected it.
- Results
Figure 1D. Multilineage differentiation of UDSCs. (a) Adipogenic differentiation; Oil red O staining for lipid droplets (Day 21). (b) Osteogenic differentiation; Alizarin red S staining for calcium deposits (Day 21). (c) Chondrogenic differentiation; Alcian blue staining for glycosaminoglycan accumulation (Day 21).
Point 6:
Could you please provide more information/explanation for Figure 1E? In the text of Figure 1E, the meaning of IRI could be written (ischemia-reperfusion acute kidney injury). Could you please add the meaning of sham (sham surgery, placebo surgery in the figure’s text?
Response 6:
We revised as below.
- Results
(Line 105) Figure 1E. PKH26-labeled UDSCs (red) were intravenously administered to wild type (Sham) and ischemia-reperfusion injury (IRI) mice.
Point 7:
Regarding Figure 2B. Could you please explain why B-actin was used in cell lysates and BSA in supernatants? Should the loading control also be present in the western blot image?
Response 7:
We used beta-actin (housekeeping protein) as a loading control to normalize protein levels in cell lysates. BSA (contained in FBS) was also used as a loading control to normalize protein levels in cell culture supernatants. The loading control was presented in the western blot image to show the equal loading of samples across all wells. If we misunderstood your point, please let us know.
Point 8:
Line 139. It may help readers not familiar with this type of research to explain that HK-2 cells are a proximal tubular cell line derived from a normal human adult male kidney; the cell line has applications in toxicology research.
Response 8:
We added as below.
- Results
(Line 140-142) HK-2 cells are a renal proximal tubular cell line derived from a normal adult kidney and have been commonly used in the in vitro study of renal cellular physiology.
Point 9:
Line 142. Instead of Figure 3A, shouldn’t be written 3B?
Response 9:
We believe that it is the right place in the sentence. Please recheck.
Point 10:
Line 150. The authors could add that “EMT is a reversible process where epithelial cells undergo defined molecular changes to become motile mesenchymal cells” to help the readers.
Response 10:
We added as below.
- Results
(Line 152-153) EMT is a reversible process where epithelial cells undergo defined molecular changes to become mesenchymal cells.
Point 11:
Lines 182–182. Could you please provide more details about the difference of canonical and non-canonical signaling pathways? Why are they relevant in this situation?
Response 11:
We added as below
- Results
(Line 183-189) TGF-β signaling pathway regulates biological features, including cell proliferation, EMT process, and ECM production through both Smad-dependent (canonical) and independent (non-canonical) ways. TGF-β/Smad pathway, the canonical pathway, regulates various fibrosis-related genes through translocation of phospho-Smad2/Smad3 which results in ECM production and MMP suppression. And TGF-β/ERK pathway, one of the non-canonical pathways, induces myofibroblast transdifferentiation and proliferation via phosphorylation of ERK[46,47].
Point 12:
Line 204. Could you please confirm that instead of “Raf,” “phosphor-c-Raf” should be written?
Response 12:
Thank you. We corrected it.
- Results
(Line 213-215) Normal UDSCs effectively downregulated ERK1/2 phosphorylation and the expression of phospho-c-Raf in HK-2 cells, as expected, whereas si-Klotho UDSC insignificantly downregulated the TGF-β/ERK pathway (Figure 6A).
Point 13:
Lines 214–217. If I am not wrong, I think that it should be stated that “the presence of Klotho proteins in UDSCs downregulated….”
Response 13:
We revised it.
- Results
(Line 226-227) Our data demonstrate that the presence of Klotho proteins in UDSCs downregulated both canonical and non-canonical pathways in the HK-2 fibrosis model.
Point 14:
Lines 259–260. I think it should be noted the result of experiment 6B in this point of the discussion.
Response 14:
In Figure 3D, 130 kDa Klotho expression was severely decreased in HK-2 cells after TGF-β1 treatment. And we observed a significant increase in 130 kDa Klotho expression of HK-2 cells after co-culture with normal UDSCs (Figure 6B). Thus, we presumed that 130 kDa full-length Klotho, detected in cell lysates, might be an HK-2-derived membrane Klotho. But we also discussed that the secretion of UDSCs-derived Klotho cannot be excluded.
Therefore, we would like to revise and add the manuscript as below.
- Results
(Line 209-210) Figure 6B. Klotho expressions were analyzed by western blotting using HK-2 cell lysates.”
- Results
(Line 217-225) We also compared the Klotho levels in HK-2 cell lysates after co-culture with Klotho siRNA-treated or untreated UDSCs. Normal UDSCs significantly increased 130 kDa Klotho expression in the HK-2 cells compared to the si-Klotho UDSCs. 130 kDa Klotho expression of HK-2 cells was relatively increased after co-culture with normal UDSCs compared to si-Klotho UDSCs. However, since the basal 65 kDa Klotho expression was low, all groups including the positive control did not show significant results. Even though there was no statistical significance, 65 kDa Klotho expression of HK-2 cells seems to be increased after co-culture with normal UDSCs (Figure 6B).
- Discussion
(Line 275-280) Conversely, normal UDSCs strongly inhibited the fibrosis of HK-2 cells. Also, we observed a significant increase in Klotho expression of HK-2 cells after co-culture with normal UDSCs. But it has not been clearly defined whether the increase of Klotho levels in HK-2 is due to the UDSCs-secreted Klotho transport or the endogenous Klotho recovery of HK-2 by itself. Further research is needed to address this aspect.
Point 15:
Could you please discuss the mechanism of Klotho expression by HK2 cells?
Response 15:
Do you mean the results of Figure 6B? (Klotho increased mechanism in HK-2 cells)
If correct, we believe ‘Respond 14’ could cover the ‘point 15’.
Point 16:
The work of Koh et al. (PMID: 11162628 DOI: 10.1006/bbrc.2000.4226) shows the lack of expression of Klotho in chronic renal failure. If figure 3 could be added to this manuscript, it will help the highlight the importance in human pathological conditions.
Response 16:
We have already mentioned in our “1. Introduction” as below.
We added your reference.
- Introduction
(Line 68-69) However, in patients with CKD, the blood concentrations of Klotho were reported to decrease as the disease progressed[37-39].
Point 17:
Could you please discuss how to increase Klotho levels and their usefulness? (demethylation of the klotho promoter, diacylation, PPAR-c antagonist, vitamin D, exogenous, etc.). Front. Endocrinol., August 27, 2020 | https://doi.org/10.3389/fendo.2020.00560
Response 17:
Yes, we added to the discussion and reference as below.
- Discussion
(Line 251-255) Klotho has been spotlighted as a potential therapeutic factor for CKD. Because of its renoprotective potential, such as fibrosis inhibition and oxidative stress reduction. In the last few years, several strategies using Klotho promoter demethylation, PPAR-γ agonist, vitamin D, and recombinant Klotho protein treatments have been reported to increase endogenous and exogenous Klotho levels[52].
Round 2
Reviewer 1 Report
I think the authors have replied to all my queries and so the manuscript can be endorsed for publication.